# Dietary Intake of Salt from Meat Products in Serbian Population

**DOI:** 10.3390/ijerph20054192

**Published:** 2023-02-26

**Authors:** Milešević Jelena, Lilić Slobodan, Vranić Danijela, Zeković Milica, Borović Branka, Glibetić Marija, Gurinović Mirjana, Milićević Dragan

**Affiliations:** 1Centre of Research Excellence in Nutrition and Metabolism, National Institute of Republic of Serbia, Institute for Medical Research, University of Belgrade, Tadeuša Košćuška, 111000 Belgrade, Serbia; 2Institute of Meat Hygiene and Technology, Kaćanskog 13, 11040 Belgrade, Serbia

**Keywords:** salt intake, sodium chloride, sodium, meat products consumption, risk assessment, estimated daily intake, CVD risk, EU Menu

## Abstract

Salt intake above 5 g/day correlates with prevalence of hypertension and cardiovascular diseases (CVD). CVD, the leading cause of mortality and morbidity in Europe, account for 45% of all deaths, while, in Serbia in 2021, CVD accounted for 47.3%. The objective was to investigate salt content labelled on meat products from the Serbian market and estimate dietary exposure to salt from meat products in the Serbian population using consumption data. Data on salt content were collected from 339 meat products and classified in eight groups. Consumption data were collected using the EFSA EU Menu methodology (2017–2021) from 576 children and 3018 adults (145 pregnant women) in four geographical regions of Serbia. The highest salt content was in dry fermented sausages and dry meat, average 3.78 ± 0.37 g/100 g and 4.40 ± 1.21 g/100 g, respectively. The average intake of meat products is 45.21 ± 39.0 g/day and estimated daily salt intake from meat products per person is 1.192 g, which is 24% of the daily recommended amount. The actual meat product consumption and content of salt in meat products in Serbia present a risk factor for development of CVD and related comorbidities. A targeted strategy, policy and legislation for salt reduction are needed.

## 1. Introduction

Salt (sodium chloride) has had great importance in the history of the human population as the first food preservative. It improved sensory attributes of food, making it shelf-stable for longer and available independent of annual season. Salt intake is influenced by many factors, such as dietary habits and culture, age, physical activity, heritage, etc. Salty taste has a hedonic dimension, and adding salt to food increases likeability up to a certain point, after which adding more salt decreases palatability [1].

Salt is the most important source of sodium, which has several roles in the organism: (i) it is a dominant cation in extracellular fluid, and, with its accompanying anion chloride, contributes extracellular osmolality of 275–295 mOsm/kg of water; (ii) it participates in control of body water volume and its systemic distribution; (iii) it enables cellular uptake of solutes and (iv) interaction with potassium in transmembrane electrochemical potentials [2].

Humans have the capacity to survive at extremely low sodium intake, less than 0.2 g per day (Yanomamo Indians from Brazil), as the body can conserve sodium by reduced loss via urine and sweat. The minimum amount of sodium required to replace losses is estimated to be no more than 0.18 g/day [3]. Salt intake above 5 g/person/day (i.e., sodium intake higher than 2 g/day) is positively correlated with average blood pressure and prevalence of hypertension and incidence of cardiovascular diseases (CVD) within the population [4].

CVD remain the leading cause of mortality and morbidity in Europe, accounting for 45% of all deaths in Europe and 37% of all deaths in the EU. In Serbia, as in previous years, in 2021, the leading causes of death (47.3%) were CVD; 46.2% of the adult population aged 20 years or over had hypertension or potential hypertension; coronary heart disease deaths in Serbia reached 21.26% of total deaths and account for the absolute leading cause of death [5], followed by stroke, both of which are related to high salt and sodium intake [6]. The age-standardized prevalence rate for CVD in Europe was 7.147, whereas, in Serbia, it was 8.274. A major medical risk factor for CVD in both genders is high blood pressure, while dietary factors make the largest contribution to risk of CVD mortality [7]. In Serbia in 2019, 9.5% of inhabitants had a habit of adding salt to food before tasting it, which did not change significantly from the 2006 and 2013 surveys [6]. Regardless, salt consumption rapidly increased with food processing. Salt production valued USD 28.5 billion in 2020 and is projected to reach a value of over USD 32 billion by 2026 [8].

Therefore, the objective of this study was to investigate the data of salt content labelled on meat products from the Serbian retail market. A further objective was to estimate dietary exposure to salt from meat products in Serbian children and the adult population using consumption data considering different ages, gender groups and regions.

## 2. Materials and Methods

### 2.1. Salt Content in Meat Products

Data on salt content declared on nutritive labels were collected from a total of 339 meat products from different groups sampled in the Serbian retail market: 39 pates, 71 cooked sausages (41 finely minced and 30 coarsely minced sausages), 66 pasteurized hams, 42 smoked meat products, 60 dry fermented sausages, 41 dry meats and 20 samples of bacon. Samples were collected in ten largest chain retail markets in Serbia, comprising 30 different Serbian and imported brands of meat products, which are commonly available in these retailers. Sampling laboratory documented all the labels collected for meat products in this exercise (by photos).

According to Serbian legislation, salt content should be determined as sodium content multiplied by 2.5. From 2004 until 2013, labelling of nutritive values was not obligatory and included content of protein, carbohydrate, fat, fibre, sodium, vitamins and minerals in 100 g/100 mL of food if food has one or more nutritive statement. From 2013 to 2017, labelling of nutritive value is mandatory and includes energy value, content of protein, carbohydrate and fat. Additionally, it can include information on content of sugars, saturated fatty acids, fibres and salt. Current Serbian legislation (Official Gazette of RS No. 19/17, 16/18, 17/20, 118/20, 17/22, 23/22 i 30/22) [9] require the food label to include the nutritional information of the energy in kJ and kcal and amounts of fat, saturated fat, carbohydrate, sugar, protein and salt in 100 g /100 mL of food, with possibility to present the nutritional value per serving or per one portion. All data are expressed with the reference to 2000 kcal of daily energy requirement, whereas salt is set to 6 g. Additionally, nutritive declaration can contain content of monounsaturated fats, polyunsaturated fats, polyols, starch, fibres, vitamins and minerals.

### 2.2. National Food Consumption Survey on Children and Adults

The National Food Consumption Survey on children and adults, including pregnant women, in compliance with the EFSA EU Menu methodology [10], was conducted between 2017 and 2022 [11,12] and included a total of 576 children and 3018 adults (145 pregnant women). Study was conducted in four geographical regions of Serbia (Belgrade, Vojvodina, Southeast Serbia and West Serbia). Data were collected using a set of questionnaires including general questionnaire, body weight and height measurements, age-appropriate Food Propensity Questionnaire (FPQ) and twice-repeated 24 h food record (for children)/dietary recall (for adults) (24HDR). The consumed portion sizes were estimated based on natural units, household measures, packaging information and a validated national Food Atlas for Portion Size Estimation [13]. For this study, average intake of meat products was assessed using data from two 24HDR only.

Data were compiled and analysed using nutritional software tool DIET ASSESS & PLAN (DAP) [14]. This study assessed consumption of meat products in children and adult population. During the dietary survey, parents reported on children’s (1–10 years old) daily consumption. Data on meat products consumption were performed according to age and gender and on the regional level. Consumed meat products were categorized in eight categories, which were defined in actual Serbian regulation on quality of meat products [15].

### 2.3. Estimation of Dietary Intake

Estimated daily intake of salt (sodium) for the following six population groups: toddlers, children, adolescents, adults, elderly and pregnant women were calculated by multiplying the mean content of salt (sodium) in each meat product category by the values taken from the distribution of the consumption amounts. Adopting the deterministic model to represent low, average and high consumers in the current study, the 25, 50, 75 and 95th percentile dietary intake was calculated for the whole Serbian population, including male and female separately. The daily intake of salt (sodium) was expressed in grams per day (g/day).

### 2.4. Statistical Analysis

Data analysis was performed using Minitab 17 Ink statistical software (Minitab Ink., Coventry, UK). The Kolmogorov–Smirnov normality test was used to check the normality of the distribution of the variables. Then, appropriate statistical tests were used for group comparisons. One-way analysis of variance (ANOVA) followed by Tukey test were used to compare differences among different consumer age groups. The level of significance was set at *p* < 0.05. The results are presented as mean ± standard deviation (SD), percentiles and ranges.

## 3. Results and Discussion

### 3.1. Salt Content in Meat Products

Salt contents in various meat product categories on the Serbian market are presented in Table 1. The lowest salt/sodium content was determined in pates (1.17 ± 0.06 g/100 g). These products are mostly sterilized, in some cases cooked or pasteurized, where a large amount of salt is not necessary to achieve the preserving effect. Pates are produced with a large amount of fat, which ensures spreadable consistence, and they have pleasant saltiness with low salt content probably due to salt perception, which is more expressible in products with high fat content, low protein content and at low pH value [16].

Cooked sausages and pasteurized ham had higher salt content than pates, around 2 g/100 g (1.90 ± 0.35 g/100 g in finely minced cooked sausages, 2.26 ± 0.40 g/100 g in coarsely minced cooked sausages and 2.10 ± 0.38 g/100 g in pasteurized ham). Higher amounts of salt in production of ham and cooked sausages are necessary for good water holding capacity and textural characteristics because these products are produced with large amounts of water as well as cooked sausages with large amounts of fat. Smoked meat products have a bit higher salt content (2.68 ± 0.53 g/100 g). They are produced from meat in pieces in which the brine is injected; afterwards, they are pasteurized in the chambers for thermal treatment while losing water, whereas the salt content becomes higher. This result is significantly lower than the results of Pleadin from 2015, who investigated salt content in meat products in Croatia and reported on average 5.34 ± 0.25% (5.16–5.51%) in smoked pork loin and 4.92 ± 0.33% (4.68–5.15%) in smoked neck [17].

The average salt content in bacon was 2.80 ± 0.83 g /100 g, but the range was very wide (2.00–5.10 g/100 g). The reason for this wide range is because bacon can be produced as a pasteurized product, where the preservative effect is based mostly on the high temperature, as well as a dried product, when it is necessary to add a larger amount of salt. Pleadin et al. (2015) reported higher average salt content of 5.09 ± 0.52 % in pasteurized bacon (4.36–6.21%) and in pasteurized speck 5.52 ± 0.91% (4.56–6.79%). In dried pancetta, the average content was 5.57 ± 0.82% (4.48–6.22%) [17].

The highest salt content was determined in dry fermented sausages and dry meat, average 3.78 ± 0.37 and 4.40 ± 1.21 g/100 g, respectively. These products are produced without thermal treatment, so low temperature and salt are the main agents against microbial spoilage. Fermentation of dry fermented sausages depends primarily on diameter of sausages and lasts for several weeks or months. The process of meat drying is long and sometimes can last even two years (dry ham with bones). In these products, the preservative effect is particularly linked to low temperature during salting or curing and to salt content, whose effect is based on decreasing water activity. The obtained results are lower than 4.14 ± 0.57% (3.02–5.32%), determined in dry sausage (narrow diameter), and then 4.37 ± 0.68% (3.34–5.48%) in kulen (wide diameter). Higher average salt content was also reported in dry meats, 6.34 ± 0.34% (5.93–7.18%) and 6.52 ± 0.54% (5.62–7.64%) for prosciutto and dry ham, respectively. The salt content in dry meat in smaller pieces was 5.46 ± 0.14% (5.36–5.56%) and 5.45 ± 0.45% (4.86–6.13%) in dry neck and dry shoulder, respectively [17].

### 3.2. Consumption of Meat Products

The most consumed meat products in all population groups are finely minced cooked sausages (88.68 g/day), coarsely minced cooked sausages (70.52 g/day) and dry fermented sausages (56.10 g/day) (Table 2). Moreover, higher consumption of bacon (45.19 g/day), smoked meat products (50.9 g/day) and pasteurized ham (36.63 g/day) is observed in the elderly population.

When looking at frequency of meat product consumption in different regions of Serbia, generally, the province of Vojvodina has the highest consumption of all types of meat products, with dominant consumption of pasteurized ham and bacon. Pasteurized ham is generally dominantly consumed in all four regions. In West Serbia, dry fermented sausages and bacon are second choice after pasteurized ham, while Southeast Serbia also significantly consumes smoked meat products. Belgrade has the lowest average consumption of meat products, even pasteurized ham (the second lowest) (Figure 1).

From the presented results on consumed amounts of meat products, it is obvious that many individual meat products’ consumptions exceed the recommended intake, which is 50–70 g/day of meat products (World Cancer Research Fund/American Institute for Cancer Research-WCRF/AICR) [18]. These consumptions might be even higher with high consumers or consumption of more than one type of meat product during the day.

Previous work on children’s exposure to nitrites through meat products in Serbia revealed that those who are consuming meat products more than 2–3 days per week (and more than 67% examinees did consume) were exceeding ADI for nitrites for 25.4% [19]. Detrimental effects of nitrites are amplified by excessive salt intake through meat products.

### 3.3. Contribution of Meat Products to Daily Salt Intake

Daily salt intake through meat products primarily depends on daily consumption of the product and its salt content. Expectedly, the average estimated daily salt intake is the lowest in toddlers (under 1 g/day) as they consume these products the least. On the other side, on the level of average consumption (P50), it was observed that the highest intake of salt was from dry fermented sausages in elderly men, who consumed 2.13 g of salt only through this meat product. Generally, this population group at average level intakes significant increments of daily salt through other meat products (bacon, dried meat, pasteurized ham, pate, smoked products), which is higher than in other population groups. Moreover, it is recorded that adolescents and adult males also intake high amounts of salt through coarsely minced cooked sausages, pate and finely minced sausages (only adult males) (Table 3). It is important to note that, in these population groups, at P50 level, consumption of individual meat products for the majority was >50 g on average, which certainly contributes to overall intake of salt (Table 2) and is quite high considering the previously mentioned recommendations from WCRF/AICR [18].

Estimated daily salt intake exceeds recommended values in the part of population that consumed a large amount of meat products (P95). Adolescent males, consuming coarsely minced cooked sausages only, exceed recommended salt intake by 13% (5.65 g of salt). Through consumption of dry meat, pregnant women exceed recommended salt intake by 9% (5.45 g of salt) and elderly males 29.6% (6.48 g of salt). Consuming dry fermented sausages, female children exceed recommended daily salt intake by 37.8%, adult males 13.4%, pregnant women 78.8% (8.94 g of salt) and elderly males 81.4% (9.07 g of salt) (Table 3). Quantities of consumed meat product at P95 were 32.1 g of pate (toddler girls) up to 250 g of coarsely minced cooked sausages (in adolescent males).

Figure 2 presents the contribution of the most important groups of meat products to dietary exposure to salt, dry meat, dry fermented sausages and finely minced cooked sausages, which contribute the most to salt exposure in all age groups.

The EFSA scientific panel considers that 2.0 g sodium/day is a safe and adequate intake (AI) for the general EU population of adults. Extrapolated adequate intakes for sodium are 0.2 g/day for infants (0–6 months) and for infants (7–12 months), 1.1 g/day for children from 1–3 years and 1.3 g/day for children from 4–6 years, 1.7 g/day for children aged 7–10 years and 2 g/day for children 11–17 years old, just as for adults [2].

Estimated mean adult salt intake (g/day) in some countries was: China 10.9, Montenegro 10.7, Portugal 10.5, Benin 9.9, Italy 9.7, India 9.1, Australia 9.0, the United States 9.0, New Zealand 8.5, Canada 8.3, England 8.1, Samoa 7.3 and Barbados 6.6 [20].

Rybicka and Nunes (2022) reported on salt intake in European countries. Average daily salt intake is: 6–7 g in Germany and Latvia; 7–8 g in Bulgaria; 8–9 g in the United Kingdom, France, Switzerland, Austria, Slovakia, the Netherlands, Denmark, Norway, Finland and Lithuania; 9–10 g in Ireland, Spain and Italy; 10–11 g in Belgium, Sweden and Estonia; 11–12 g in Poland and Romania; 12–13 g in Slovenia and Portugal; 13–14 g in the Czech Republic and 14–15 g in Hungary [21]. Another review on salt intake in Euro-Asian region from 2022 reports on much higher salt intake, especially in south-eastern Europe, in comparison to western Europe. In the review, it was reported that daily intake of salt in Serbia is 10.01 g/day [22,23]. This information is, however, based on the earlier study (from 2016) on salt intake of students in the province of Vojvodina. A new comprehensive study on dietary intake of salt in complete diet in Serbia is needed.

Based on some investigations, through 100 g of meat and fish product, sodium intake exceeds over 47% of the WHO recommended daily maximum in China and in the USA, in South Africa 37%, in Australia 35% and 27% in the UK. Song et al. (2021) reported that the WHO recommendation was surpassed for one third or one half with a 100 g serving size of meat and fish products [24]. In Croatia, in 2019, it was determined that 1.95 g/day of salt in daily intake was from meat products, which accounted for 39% of daily recommendations [25]. More precisely, average daily salt intake from meat products in some parts of Croatia was 2.16 g/day in Istra and Dalmacija, 2.96 g/day in middle parts of Croatia and 3.01 g/day in eastern parts of Croatia [17]. These figures resonate with the presented findings from our work. We assessed that average intake of all meat products is 45.21 ± 39.0 g/day and estimated daily salt intake from all meat products per person is 1192 mg of salt, which is 24% of daily recommended amount of salt, coming only from meat products (Table 4).

Considering previously referenced recommendations, we can conclude that the actual meat product consumption and content of salt in meat products in Serbia present a risk factor for development of various non-communicable chronic diseases, i.e., cardiovascular diseases and related comorbidities, and require various activities and initiatives for salt reduction on different levels in society. Another aspect of the study reveals the need to invest in obtaining analytical values and monitoring of salt and sodium in meat products. These data will be used for update of Serbian food composition database [26] and for further exposure assessments and monitoring of salt in food products.

Education of consumers on various aspects of sustainable healthy diet, including choosing (meat) products with less salt, which will reduce the risk of diet-related NCDs and simultaneously have effects on reduction in carbon footprint and preservation of biodiversity, is another necessary element that is needed in establishment of national food-based dietary guidelines and an overall sustainable and healthy food environment [27].

### 3.4. Salt Reduction Strategies

The Global Panel on Agriculture and Food Systems for Nutrition calls for a transformation of food systems, which will ensure sustainable, healthy diets and conversion of the food environment in a way that promotes greater diversity, availability, affordability and safety of nutritious foods. Improving nutrition through enhanced food environments also includes reformulation, labelling and processing foods in ways that increase their nutritional value and safety [28]. There are several international and European recommendations and policy actions/strategies for salt and sodium intake reduction to achieve the Global Nutrition Targets and diet-related non-communicable disease (NCD) voluntary targets [29,30,31]. However, it is essential that these policy recommendations are adapted and implemented through various legislative and technological measures and promoted widely among food system stakeholders—government, food producers and retailers, researchers, the health sector and consumers [28]. In 2020, the WHO Regional Office for Europe issued a technical document “Accelerating Salt Reduction In Europe: A Country Support Package To Reduce Population Salt Intake In The Who European Region”, which provided a clear step-wise approach guide to address salt reduction programs, including building and managing teams of stakeholders that are going to work on setting up targets and developing plans, technical support and examples of policy interventions [32].

A recent review observing how much actually has been accomplished on a national level regarding salt-reduction initiatives in the period of 2014 to 2019 revealed that, despite the increase in the number of countries adopting strategies to reduce their population-level salt intake, none have yet met the targeted 30% relative reduction in salt intake, which was set to be achieved by 2025. Most applied approaches included interventions in settings (*n* = 74 countries), food reformulation (*n* = 68 countries), consumer education (*n* = 50 countries), front-of-pack labelling (*n* = 48 countries) and salt taxation (*n* = 5 countries). More action and effort are needed to ensure that countries implement, monitor and evaluate their strategies [33]. As we see in the national program for prevention, treatment and control of cardiovascular diseases in the Republic of Serbia, until 2020 (from 2010), we foresaw that there should be adaptations, with expert recommendations, regarding salt, sugar and fat reduction in processed foods [34]. These quotes without concrete action plans have a limited impact on improvements in the field.

Moderate salt reduction could be achieved by reducing salt content in food by producers as well as a continuous campaign in the media. EHN points out that population salt reduction must be reached gradually to allow adaptation of taste preference and ensure consumer acceptance [35].

A recent study in China modelled that reducing salt intake even by a modest amount of just 1 g/day might prevent some 9 million CVD events (of which almost 4 million are fatal) if sustained until 2030 [36]. The WHO estimates that 2.5 million deaths could be prevented each year by reducing salt content in food to the recommended level [34].

Voluntary reformulation initiatives and other available measures, such as mandatory warning labels and/or legislated maximum limits for salt, are the key actions to reduce salt intake [34]. Thus, the WHO engaged with world food industry representatives in setting sodium benchmarks for particular food categories, which are useful guides for development of national recommendations, policies and legislation [37].

Particularly in meat products, a study by Lilic et al. (2016) reported that replacement of salt with 30% potassium chloride and 12.5% ammonium chloride did not have a major influence on saltiness and overall sensory characteristics of dry fermented sausages, while sausages with 50% potassium chloride replacement had expressible bitter taste [38]. Moreover, partial replacement of salt with one third of potassium chloride or ammonium chloride did not affect sensory attributes of dry pork, while larger amounts of these salts led to appearance of bitter taste [39].

Introduction of clear and consistent front-of-pack nutrition labelling helps consumers to make informed decisions about the food they are buying [40]. By raising awareness, demand toward less salty products will increase and push the food industry toward reformulation. Special focus in consumer education should be on vulnerable populations, such as socioeconomically disadvantaged groups where health literacy is often lower, while total salt intake may be up to 5–10% higher due to addition of salt at the table [41].

It has become clear that a multi-causal problem, such as high salt intake from food, cannot be addressed with scattered unlinked interventions but with targeted mutually supportive integrated approaches that are embedded within wider policy frameworks. Many governments might feel that they lack capacity to design and manage policies or enforce them on the food industry and hold them accountable. Involvement of stakeholders is necessary [34].

A positive example of this is presented in a study conducted in Bosnia and Herzegovina (B&H). The country already had a strategy for salt reduction in place and positive results in implementation of warning signs on salt packages were reported. The measure was cost-effective in terms of application (printing labels with warnings), while it informed consumers and caused a decrease in salt intake by 13% in 12 months of release, concomitantly reducing blood pressure [42]. Aside from this population approach, the B&H salt reduction strategy took the stakeholder approach, which entailed industry involvement, i.e., food reformulation targeting specific food categories [43].

## 4. Conclusions

This study, for the first time, examines actual dietary intake and exposure to salt from meat products in Serbia. The salt intake but also amount of meat products consumed daily among Serbian adults raise a concern of potential NCD risk, correlating with previous findings on high intake of salt, high rate of hypertension and other CVD incidence. Serbia is a country without a targeted strategy for salt reduction even though this issue has been considered in the national program for prevention of obesity among children and adults (from 2018). The program places salt reduction among the key goals to be achieved until 2025 and recognizes that there is a need to work closely with the food industry and on development of action plans for implementation of mechanisms that will induce reduction in content of salt (among other harmful substances in food). This particularly entails improving legislation on nutritional labelling, establishment of communication and lobbying the food industry. This concept can and should be enlarged and populated with various other elements, including taxation of HFSS foods, regulation of allowed content of salt (nutritional profiling) and raising awareness among consumers using different forms of communication. Further research should focus on determination of salt intake from comprehensive food consumption of children and adults to provide solid evidence for policymaking.

## Figures and Tables

**Figure 1 ijerph-20-04192-f001:**
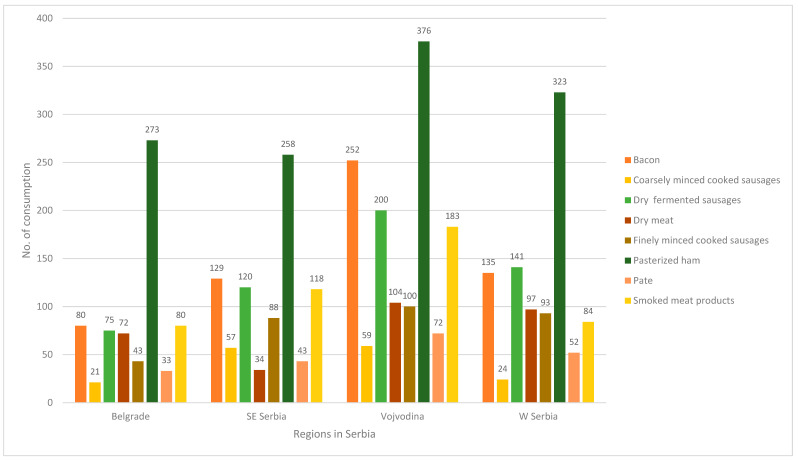
Frequency of consumption of meat products in four regions in Serbia.

**Figure 2 ijerph-20-04192-f002:**
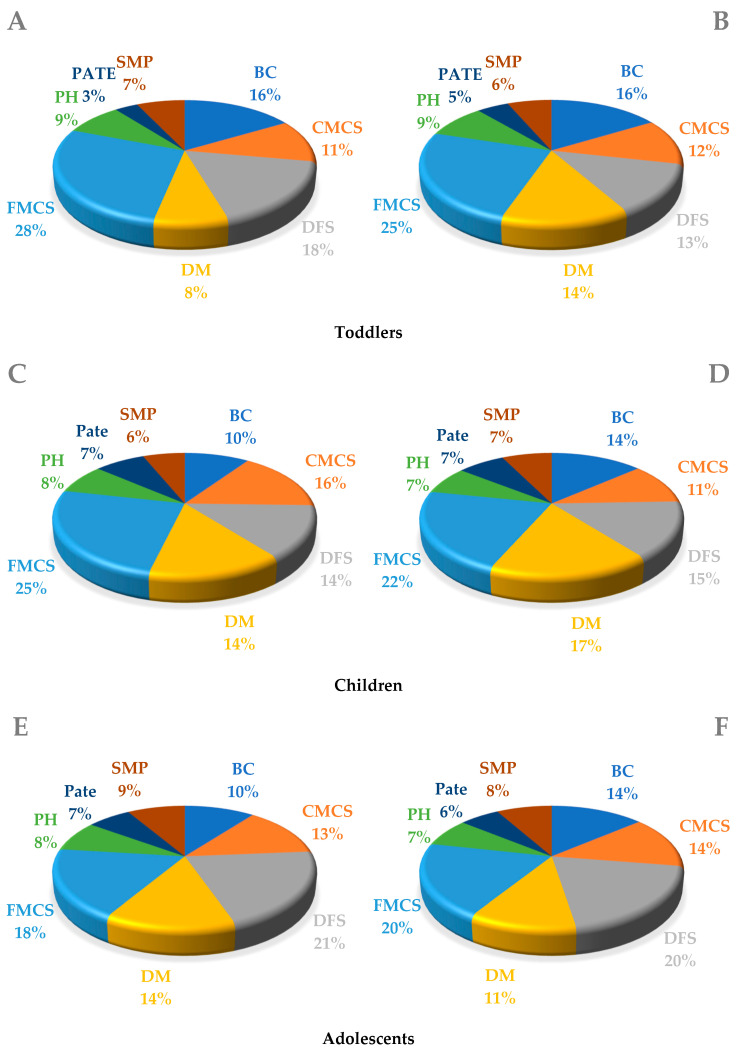
Contribution (%) of the most important groups of meat products to dietary exposure to salt. Legend: CMCS—coarsely minced cooked sausages; DFS—dry fermented sausages; DM—dry meat; FMCS—finely minced cooked sausages; PH—pasteurized ham; SMP—smoked meat products; (**A**,**C**,**E**,**G**,**I**) male; (**B**,**D**,**F**,**H**,**J)** female; (**K**) pregnant women.

**Table 1 ijerph-20-04192-t001:** Salt (sodium) content in meat products, g/100 g.

	*N*	Mean ± SD	P25	P50	P75	P95	Range
Thermally treated products							
Pate	39	1.17 ± 0.06 (0.47 ± 0.03)	1.10 (0.44)	1.20 (0.48)	1.20 (0.48)	1.30 (0.52)	1.10–1.30 (0.44–0.52)
Finely minced cooked sausages	41	1.90 ± 0.35 (0.76 ± 0.14)	1.60 (0.64)	1.90 (0.76)	2.10 (0.84)	2.60 (1.04)	1.40–2.60 (0.56–1.04)
Coarsely minced cooked sausages	30	2.26 ± 0.40 (0.91 ± 0.16)	2.00 (0.80)	2.20 (0.88)	2.40 (0.96)	3.30 (1.32)	1.70–3.30 (0.68–1.32)
Pasteurized ham	66	2.10 ± 0.38 (0.84 ± 0.15)	1.80 (0.72)	2.00 (0.80)	2.50 (1.00)	2.70 (1.08)	1.20–2.85 (0.48–1.14)
Smoked meat products	42	2.68 ± 0.53 (1.07 ± 0.21)	2.22 (0.89)	2.50 (1.00)	3.30 (1.32)	3.30 (1.32)	1.70–3.30 (0.68–1.32)
Thermally untreated products							
Dry fermented sausages	60	3.78 ± 0.37 (1.51 ± 0.15)	3.60 (1.44)	3.80 (1.52)	4.0 (1.60)	4.50 (1.80)	3.10–4.50 (1.24–1.80)
Dry meat	41	4.40 ± 1.21 (1.76 ± 0.49)	3.60 (1.44)	4.30 (1.72)	5.30 (2.12)	6.40 (2.56)	2.60–6.40 (1.04–2.56)
Bacon	20	2.80 ± 0.83 (0.11 ± 0.33)	2.40 (0.96)	2.55 (1.02)	2.80 (1.12)	5.10 (2.04)	2.00–5.10 (0.80–2.04)

*N*—number of analysed samples.

**Table 2 ijerph-20-04192-t002:** Distribution of consumption (g/day) of meat products by populations (age and gender).

Meat Products		Daily Consumption of Meat Products (g/Day)										
		Age (Years) and Gender										
		Toddlers (1–3 y)		Children (3–9 y)		Adolescents (10–17 y)		Adults (18–64 y)		Elderly (64–74 y)		Pregnant Women
		M	F	M	F	M	F	M	F	M	F	
B	P25	20.00	21.73	13.34	20.17	20	24.25	21.50	15.35	30	22	10.66
	P50	22.00	24.28	22.00	26.50	33.68	39.085	42.66	31.00	47	31	25
	P75	30.00	42.74	36.16	43.93	51.695	56.88	58.39	50.00	62.71	50	48.5
	P95	50.02	116.20	93.64	53.11	100	97.3766	120.51	99.01	103.31	100	71.66
CMCS	P25	17	20	27.5	20	31.83	30	50	30	38	40	35
	P50	27.5	30	36	30	100	40	100	50	50	50	50
	P75	47.5	41	40	45	150	100	140	75	56.25	75	100
	P95	111.2	46.85	42.75	48	250	192	150	169	150	150	100
DFS	P25	15.79	13.00	14.45	16.5	30	26	30	26	37.76	20	30.00
	P50	22.95	30.00	28	26	42	30	50	42	56.44	33	45.08
	P75	31.35	40.45	41.95	41.91	62.71	50	84	53.72	96.61	55.18	53.89
	P95	49.59	49.59	55.79	182.4	113.51	100	150	120	240	120	236.40
DM	P25	6.07	11.50	12.74	16.25	17.09	12.82	25.64	17.06	30	23.18	34.59
	P50	12.50	19.20	22.5	30	30	25	37.5	30	40	30	50
	P75	23.55	22.50	30	52.58	50	50	50	46.87	50	43.39	72.73
	P95	25.45	30.00	48	96.4	89.2	100	100	80.8	147.2	86.74	123.96
FMCS	P25	49.38	48.13	51	47.5	52.25	50.75	80	50.37	73.18	50.25	46.77
	P50	51.00	51.00	84	80	84	84	100	84	84	81.9	72.5
	P75	72.50	52.25	93.29	96.9	104.5	100	159.6	100	112.28	100	137.5
	P95	108.72	152.28	145.24	106.4	192.32	169.48	252	163.10	212.32	168.28	152
PH	P25	13.83	15.815	14.58	14.58	20.80	17.50	22.5	16	22.81	18.56	15.72
	P50	16	16.67	20	20	24.44	22.92	30	22.92	37	27.78	21.87
	P75	30	20.83	30	34	33.71	32.00	40	34	56.05	34.62	35.69
	P95	50	80.95	50	50	62.70	50.00	80	60.76	102.5	95.95	68.1
P	P25	10	15	25	25	30	25	48.27	27.5	37.5	30	28.75
	P50	25	25	30	25	50	40	50	50	50	40	42.5
	P75	25	25	40	50	75	50	75	60	75	50	50
	P95	43.7	32.1	50	79.2	95.75	54.5	109	100	100	75	67.5
SMP	P25	9	9	9	11.41	17.09	15	20	17.54	25.25	20	17.09
	P50	20	20	18	16	25.51	29.09	30	30	50	27	24.99
	P75	45	21.25	30	32.54	46.25	45	60	50	100	41	47.5
	P95	89	36	100	96	80	74.8	100	122.4	120	100	140

Legend: B—bacon; CMCS—coarsely minced cooked sausages; DFS—dry fermented sausages; DM—dry meat; FMCS—finely minced cooked sausages; PH—pasteurized ham; P—pate; SMP—smoked meat products.

**Table 3 ijerph-20-04192-t003:** Distribution of daily intake of salt (sodium) (g/day) through consumption of meat products.

Meat Products		Estimated Daily Intake of Salt (Sodium) (g/Day)										
		Age (Years) and Gender										
		Toddlers (1–3 y)		Children (3–9 y)		Adolescents (10–17 y)		Adults (18–64 y)		Elderly (64–74 y)		Pregnant Women
		M	F	M	F	M	F	M	F	M	F	
B	P25	0.56 (0.224)	0.61 (0.243)	0.37 (0.149)	0.56 (0.226)	0.56 (0.224)	0.68 (0.272)	0.60 (0.241)	0.43 (0.172)	0.84 (0.336)	0.62 (0.246)	0.30 (0.119)
	P50	0.62 (0.246)	0.68 (0.271)	0.62 (0.246)	0.74 (0.297)	0.94 (0.377)	1.09 (0.438)	1.19 (0.478)	0.87 (0.347)	1.32 (0.526)	0.87 (0.347)	0.70 (0.280)
	P75	0.84 (0.336)	1.20 (0.478)	1.01 (0.405)	1.23 (0.492)	1.45 (0.529)	1.59 (0.637)	1.63 (0.654)	1.40 (0.560)	1.76 (0.702)	1.40 (0.560)	1.36 (0.543)
	P95	1.40 (0.560)	3.25 (1.301)	2.62 (1.049)	1.49 (0.595)	2.80 (1.120)	2.73 (1.091)	3.37 (1.350)	2.77 (1.109)	2.89 (1.157)	2.80 (1.120)	2.01 (0.803)
CMCS	P25	0.38 (0.154)	0.45 (0.181)	0.62 (0.249)	0.45 (0.181)	0.72 (0.290)	0.68 (0.272)	1.13 (0.453)	0.68 (0.272)	0.86 (0.345)	0.90 (0.363)	0.79 (0.317)
	P50	0.62 (0.249	0.68 (0.272)	0.81 (0.326)	0.68 (0.272)	2.26 (0.907)	0.90 (0.363)	2.26 (0.907)	1.13 (0.453)	1.13 (0.453)	1.13 (0.453)	1.13 (0.453)
	P75	1.07 (0.431)	0.93 (0.372)	0.90 (0.363)	1.02 (0.408)	3.39 (1.360)	2.26 (0.907)	3.16 (1.270)	1.69 (0.680)	1.27 (0.510)	1.70 (0.680)	2.26 (0.907)
	P95	2.51 (1.01)	1.06 (0.425)	0.97 (0.390)	1.08 (0.435)	5.65 (2.267)	4.34 (1.741)	3.39 (1.538)	3.82 (1.533)	3.39 (1.360)	3.39 (1.360)	2.26 (0.907)
DFS	P25	0.60 (0.239)	0.49 (0.197)	0.55 (0.219)	0.62 (0.250)	1.13 (0.454)	0.98 (0.393)	1.13 (0.454)	0.98 (0.393)	1.43 (0.571)	0.76 (0.302)	1.13 (0.454)
	P50	0.87 (0.347)	1.13 (0.454)	1.06 (0.424)	0.98 (0.393)	1.59 (0.635)	1.13 (0.454)	1.89 (0.756)	1.59 (0.635)	2.13 (0.854)	1.25 (0.500)	1.70 (0.682)
	P75	1.19 (0.474)	1.53 (0.612)	1.59 (0.635)	1.58 (0.643)	2.37 (0.950)	1.89 (0.756)	3.18 (1.271)	2.03 (0.813)	3.65 (1.462)	2.09 (0.835)	2.04 (0.815)
	P95	1.87 (0.750)	1.87 (0.750)	2.11 (0.844)	6.89 (2.760)	4.29 (1.717)	3.78 (1.513)	5.67 (2.270)	4.54 (1.816)	9.07 (3.631)	4.54 (1.816)	8.94 (3.577)
DM	P25	0.27 (0.106)	0.51 (0.202)	0.56 (0.224)	0.72 (0.285)	0.75 (0.300)	0.56 (0.225)	1.13 (0.450)	0.75 (0.300)	1.32 (0.527)	1.02 (0.407)	1.52 (0.610)
	P50	0.55 (0.220)	0.84 (0.337)	0.99 (0.395)	1.32 (0.527)	1.32 (0.527)	1.10 (0.440)	1.65 (0.660)	1.32 (0.527)	1.76 (0.703)	1.32 (0.527)	2.20 (0.880)
	P75	1.04 (0.414)	0.99 (0.395)	1.32 (0.527)	2.31 (0.924)	2.20 (0.880)	2.20 (0.880)	2.20 (0.880)	2.06 (0.823)	2.20 (0.880)	1.91 (0.762)	3.20 (1.280)
	P95	1.12 (0.447)	1.32 (0.527(	2.11 (0.843)	4.24 (1.694)	3.92 (1.567)	4.40 (1.757)	4.40 (1.757)	3.56 (1.420)	6.48 (2.600)	3.82 (1.524)	5.45 (2.180)
FMCS	P25	0.94 (0.375)	0.91 (0.366)	0.97 (0.390)	0.90 (0.361)	0.99 (0.400)	0.96 (0.386)	1.52 (0.610)	0.96 (0.383)	1.39 (0.557)	0.95 (0.382)	0.89 (0.356)
	P50	0.97 (0.390)	0.97 (0.390)	1.60 (0.640)	1.52 (0.610)	1.60 (0.640)	1.60 (0.640)	1.90 (0.761)	1.60 (0.640)	1.60 (0.640)	1.56 (0.623)	1.38 (0.552)
	P75	1.38 (0.552)	0.99 (0.400)	1.77 (0.710)	1.84 (0.740)	1.99 (0.795)	1.90 (0.761)	3.03 (1.215)	1.90 (0.761)	2.13 (0.854)	1.90 (0.761)	2.61 (1.046)
	P95	2.07 (0.830)	2.89 (1.160)	2.76 (1.105)	2.02 (0.810)	3.65 (1.464)	3.22 (1.290)	4.79 (1.920)	3.10 (1.241)	4.03 (1.616)	3.20 (1.281)	2.89 (1.157)
PH	P25	0.29 (0.117)	0.33 (0.133)	0.31 (0.122)	0.31 (0.122)	0.44 (0.175)	0.37 (0.147)	0.47 (0.190)	0.34 (0.134)	0.48 (0.192)	0.39 (0.156)	0.33 (0.132)
	P50	0.34 (0.134)	0.35 (0.140)	0.42 (0.168)	0.42 (0.168)	0.51 (0.205)	0.48 (0.192)	0.63 (0.252)	0.48 (0.193)	0.78 (0.311)	0.58 (0.233)	0.46 (0.184)
	P75	0.63 (0.252)	0.44 (0.175)	0.63 (0.252)	0.71 (0.286)	0.71 (0.283)	0.67 (0.270)	0.84 (0.336)	0.71 (0.286)	1.18 (0.471)	0.73 (0.291)	0.75 (0.300)
	P95	1.05 (0.420)	1.70 (0.680)	1.05 (0.420)	1.05 (0.420)	1.32 (0.527)	1.05 (0.420)	1.68 (0.672)	1.28 (0.510)	2.15 (0.861)	2.01 (0.806)	1.43 (0.572)
P	P25	0.12 (0.05)	0.18 (0.070)	0.29 (0.117)	0.29 (0.117)	0.35 (0.141)	0.29 (0.117)	0.56 (0.226)	0.32 (0.130)	0.44 (0.176)	0.35 (0.141)	0.34 (0.135)
	P50	0.29 (0.117)	0.29 (0.117)	0.35 (0.141)	0.29 (0.117)	0.59 (0.234)	0.47 (0.190)	0.59 (0.234)	0.59 (0.234)	0.59 (0.234)	0.47 (0.190)	0.50 (0.200)
	P75	0.29 (0.117)	0.29 (0.117)	0.47 (0.188)	0.59 (0.234)	0.88 (0.352)	0.59 (0.234)	0.88 (0.352)	0.70 (0.281)	0.88 (0.352)	0.59 (0.234)	0.59 (0.234)
	P95	0.51 (0.205)	0.38 (0.151)	0.59 (0.234)	0.93 (0.371)	1.12 (0.450)	0.64 (0.256)	1.28 (0.511)	1.17 (0.470)	1.17 (0.470)	0.88 (0.352)	0.79 (0.317)
SMP	P25	0.24 (0.100)	0.24 (0.100)	0.24 (0.100)	0.31 (0.122)	0.46 (0.183)	0.40 (0.161)	0.54 (0.214)	0.47 (0.190)	0.68 (0.270)	0.54 (0.214)	0.46 (0.183)
	P50	0.54 (0.214)	0.54 (0.214)	0.48 (0.193)	0.43 (0.171)	0.68 (0.273)	0.78 (0.312)	0.80 (0.321)	0.80 (0.321)	1.34 (0.535)	0.72 (0.290)	0.67 (0.270)
	P75	1.21 (0.482)	0.57 (0.230)	0.80 (0.321)	0.87 (0.350)	1.24 (0.500)	1.21 (0.482)	1.61 (0.643)	1.34 (0.535)	2.68 (1.071)	1.10 (0.440)	1.27 (0.510)
	P95	2.39 (0.953)	0.96 (0.390)	2.68 (1.071)	2.57 (1.030)	2.14 (0.860)	2.00 (0.801)	2.68 (1.071)	3.28 (1.311)	3.22 (1.285)	2.68 (1.071)	3.75 (1.500)

Legend: B—bacon; CMCS—coarsely minced cooked sausages; DFS—dry fermented sausages; DM—dry meat; FMCS—finely minced cooked sausages; PH—pasteurized ham; P—pate; SMP—smoked meat products.

**Table 4 ijerph-20-04192-t004:** Average consumption of meat products and estimated daily intake of salt from meat products per age and gender group.

Age Groups	Average Consumption of Meat Products (g/Day)												$\overset{-}{x}$ (g/100 g) ^1^	Estimated Daily Intake of Salt (g/Day)		
	F				M				Total							
	Mean ± Sd		Min.	Max.	Mean ± Sd		Min.	Max.	Mean ± Sd		Min.	Max.		F	M	Total
Toddlers	31.65	24.72	7.50	154.55	32.82	25.44	0.12	150.00	32.24	25.05	0.12	154.55	2.63	0.834	0.865	0.850
Children	34.59 ^c^	27.81	2.47	240.00	34.18 ^c^	27.01	4.10	168.00	34.38 ^a^	27.38	2.47	240		0.912	0.901	0.906
Adolescents	40.63 ^b.c^	36.17	0.17	300.00	44.68 ^b^	37.80	0.12	300.00	42.96 ^b^	37.16	0.12	300		1.071	1.178	1.133
Adults	40.50 ^b.c^	33.81	0.17	250.00	54.14 ^a^	45.20	0.25	400.00	48.99 ^c^	41.78	0.17	400		1.068	1.427	1.292
Elderly	44.52 ^b^	35.98	4.00	240.00	61.23 ^a^	48.53	2.47	360.00	52.75 ^c^	43.41	2.47	360		1.174	1.614	1.391
Pregnant women	42.39	36.66	0.30	240.00					42.39	36.66	0.3	240		1.118	0.000	1.118
Total	40.26	34.21	0.17	300.00	49.29	42.11	0.12	400.00	45.21	39.00	0.12	400		1.061	1.299	1.192

$\overset{-}{x}$ ^1^ Average salt content in meat products (g/100 g); ^a–c^—means that share a similar letter in the same column are not significantly different (*p* ≤ 0.05).

## Data Availability

Not applicable.

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
