# Peer review of "Dietary Intake of Salt from Meat Products in Serbian Population"

_ijerph, 2023, doi:10.3390/ijerph20054192_

Round 1

Reviewer 1 Report

This manuscript is about dietary intake of salt from meat products and  the risk assessment in a Serbian Population

The manuscript is well written and presented

However, I have some remarks

How did the authors obtained informations from children? Did they ask their parents ?

Results and discussion

-The authors should introduce the results of their research before discussing the universal dietary sources of salt and its intake in the world context. They should compare their results with its intake in the word 

-The survey should be added in this manuscript 

Author Response

Dear Editor and reviewer,

We sincerely appreciate your insightful and valuable comments and suggestions. All of your comments and suggestions are of great help to the improvement of the quality of our manuscript. Here we submit a new version of our revised manuscript according to the suggestions.

The manuscript is now substantially shortened, list of references is reduced, and Title survived a bit of change.

Please find detailed response for all three Reviewers.

(Reviewer 1)

The manuscript is well written and presented

However, I have some remarks

How did the authors obtained informations from children? Did they ask their parents ?

Thank you for the question. During the dietary survey, parents reported on children (1-10 years old) daily consumption.

Results and discussion

-The authors should introduce the results of their research before discussing the universal dietary sources of salt and its intake in the world context. They should compare their results with its intake in the word

Thank You for the suggestion. We have moved the results to the beginning of the section.

-The survey should be added in this manuscript 

The questionnaires used for the dietary survey are in Serbian language and are under protection of Intellectual property rights.

(Reviewer 2)

The authors present interesting and relevant data related to salt content of meat products in Serbia and contribution to daily intake. As it is, it reads like a thesis or parts of. It is very lengthy and many parts could be reduced considerably to adhere to a style of a manuscript. There are over 100 references! Which also speaks to the very detailed background information included, which in many parts is not necessary to link directly back to the rationale and aim of the study. Further comments below

Thank you for the suggestion. Many sections in the introduction are now deleted and reduced to make reading concise.

Abstract – is there an error in the reporting of the value of 1,192 grams of salt e.g.  ? decimal place needed here rather than comma 1.192 g ?

Thank you. Amended.

The introduction is very lengthy, I would suggest shortening it. IT could be much more concise in many areas e.g. too many stats on CVD and HBP – these could be reduced. Too much detail on all the other associated health conditions with high salt – it seems like the focus and rationale for the article can be linked back to CVD alone. A few of these paragraphs could be deleted.

Thank you for the suggestion. Many sections in the introduction are now deleted and reduced to make reading concise.

Intro – can a reference be provided for statement that high fat foods contain more salt than those of higher protein content.

Thank you. The sentence, lines 113-115 are deleted.

Likewise, the paragraph on technical role of salt in food is lengthy in description and could be reduced.

Methods – were actual hard copy labels of nutrition information panels checked? Or online sources used? How did you decide what retail outlets to sample in Serbia – can you provide some more information on sampling here.

Thank You. We provided detailed explanation the in the section 2.1. 

“Samples were collected in ten largest chain retail markets in Serbia, comprising 30 different Serbian and imported brands of meat products, which are commonly available in these retails. Sampling laboratory documented all the labels collected for meat products in this exercise (by photos).”

Results and discussion: this starts of with a lot of detail about sources of salt from foods across many different countries before any data from the present result is presented. I understand the journal allows combination of results and discussion but I would recommend in first instance your own study results are clearly presented; then discuss these in the context of the other relevant findings e.g. section 3.1 needs to be removed/reworked elsewhere

Thank you. We did reduce and reorganized the whole results.

The paragraph from line 264, page 6 seems out of context (E.g. methods to determine Na content). Could it not be a simple mention in your limitations section that determining sodium from reported food labels is a limitation in this method and lab based methods such as xxx (summarise) are recommended/needed to confirm findings.

Thank You. Lines from 264 up to 277 deleted.

 Methods – did you take the average of the 2 days of 24hr recall data to collate the information on meat consumption? There is mention of another frequency dietary questionnaire – was this used at all. If so please clarify further what/how the dietary data in the main study was used in your analysis.

For this study, average intake of meat products was assessed using data from two 24HDR only.

Section 3.4 in results is very long, multiple paragraphs describing sodium intakes in different areas. Pick the key areas/population groups to compare your findings to. This section could be reduced significantly

Thank you. We did reduce and reorganized the whole results.

Section 3.5 – while interesting it is incredibly long and ? how much of it is needed to directly talk about the results presented in this study.

Thank you. We did reduce and reorganized the whole results.

(Reviewer 3)

The manus is interesting and important; however, it is very dense, which hinders the reading a lot. I would immediately suggest a mandatory modification and that is to separate the results section from the discussion of the article. This is relevant and will significantly transform the article to a better reading which is not possible in its present form.

Thank you. We did reduce and reorganized the whole results.

 The tables also have a lot of information, and reading will be very beneficial if, instead of putting the salt and sodium values in all the Tables, authors describe only the salt values since sodium derives from it.

Thank you. Since we are referencing to WHO (salt) and EFSA (sodium) we prefer to leave both values for better overview for a reader who seek deeper info.

 Regarding other methodological issues, I do not find anything relevant that I can add to help the authors. Only the title can be simplified to "Dietary intake of salt from meat products in a Serbian population"

We agreed. Title is now changed.

Reviewer 2 Report

The authors present interesting and relevant data related to salt content of meat products in Serbia and contribution to daily intake. As it is, it reads like a thesis or parts of. It is very lengthy and many parts could be reduced considerably to adhere to a style of a manuscript. There are over 100 references! Which also speaks to the very detailed background information included, which in many parts is not necessary to link directly back to the rationale and aim of the study. Further comments below

·       Abstract – is there an error in the reporting of the value of 1,192 grams of salt e.g.  ? decimal place needed here rather than comma 1.192 g ?

·       The introduction is very lengthy, I would suggest shortening it. IT could be much more concise in many areas e.g. too many stats on CVD and HBP – these could be reduced. Too much detail on all the other associated health conditions with high salt – it seems like the focus and rationale for the article can be linked back to CVD alone. A few of these paragraphs could be deleted.

·       Intro – can a reference be provided for statement that high fat foods contain more salt than those of higher protein content

·       Likewise, the paragraph on technical role of salt in food is lengthy in description and could be reduced.

·       Methods – were actual hard copy labels of nutrition information panels checked? Or online sources used? How did you decide what retail outlets to sample in Serbia – can you provide some more information on sampling here

·       Results and discussion: this starts of with a lot of detail about sources of salt from foods across many different countries before any data from the present result is presented. I understand the journal allows combination of results and discussion but I would recommend in first instance your own study results are clearly presented; then discuss these in the context of the other relevant findings e.g. section 3.1 needs to be removed/reworked elsewhere

·       The paragraph from line 264, page 6 seems out of context (E.g. methods to determine Na content). Could it not be a simple mention in your limitations section that determining sodium from reported food labels is a limitation in this method and lab based methods such as xxx (summarise) are recommended/needed to confirm findings

·       Methods – did you take the average of the 2 days of 24hr recall data to collate the information on meat consumption? There is mention of another frequency dietary questionnaire – was this used at all. If so please clarify further what/how the dietary data in the main study was used in your analysis

·       Section 3.4 in results is very long, multiple paragraphs describing sodium intakes in different areas. Pick the key areas/population groups to compare your findings to. This section could be reduced significantly

·       Section 3.5 – while interesting it is incredibly long and ? how much of it is needed to directly talk about the results presented in this study.

·        

Author Response

(The authors gave the same response as above.)

Reviewer 3 Report

The manus is interesting and important; however, it is very dense, which hinders the reading a lot. I would immediately suggest a mandatory modification and that is to separate the results section from the discussion of the article. This is relevant and will significantly transform the article to a better reading which is not possible in its present form.

The tables also have a lot of information, and reading will be very beneficial if, instead of putting the salt and sodium values in all the Tables, authors describe only the salt values since sodium derives from it.

Regarding other methodological issues, I do not find anything relevant that I can add to help the authors. Only the title can be simplified to "Dietary intake of salt from meat products in a Serbian population"

Author Response

Dear Editor and reviewer,

We sincerely appreciate your insightful and valuable comments and suggestions. All of your comments and suggestions are of great help to the improvement of the quality of our manuscript. Here we submit a new version of our revised manuscript according to the suggestions.

The manuscript is now substantially shortened, the list of references is reduced, and the Title survived a bit of change.

Please find the detailed responses for all three Reviewers.

(Reviewer 1)

The manuscript is well-written and presented

However, I have some remarks

How did the authors obtain information from children? Did they ask their parents?

Thank you for the question. During the dietary survey, parents reported on children's (1-10 years old) daily consumption.

Results and discussion

-The authors should introduce the results of their research before discussing the universal dietary sources of salt and its intake in the world context. They should compare their results with their intake in the word

Thank You for the suggestion. We have moved the results to the beginning of the section.

-The survey should be added to this manuscript 

The questionnaires used for the dietary survey are in Serbian language and are under the protection of Intellectual property rights.

(Reviewer 2)

The authors present interesting and relevant data related to salt content of meat products in Serbia and contribution to daily intake. As it is, it reads like a thesis or parts of. It is very lengthy and many parts could be reduced considerably to adhere to a style of a manuscript. There are over 100 references! Which also speaks to the very detailed background information included, which in many parts is not necessary to link directly back to the rationale and aim of the study. Further comments below

Thank you for the suggestion. Many sections in the introduction are now deleted and reduced to make reading concise.

Abstract – is there an error in the reporting of the value of 1,192 grams of salt e.g.  ? decimal place needed here rather than comma 1.192 g ?

Thank you. Amended.

The introduction is very lengthy, I would suggest shortening it. IT could be much more concise in many areas e.g. too many stats on CVD and HBP – these could be reduced. Too much detail on all the other associated health conditions with high salt – it seems like the focus and rationale for the article can be linked back to CVD alone. A few of these paragraphs could be deleted.

Thank you for the suggestion. Many sections in the introduction are now deleted and reduced to make reading concise.

Intro – can a reference be provided for statement that high fat foods contain more salt than those of higher protein content.

Thank you. The sentence, lines 113-115 are deleted.

Likewise, the paragraph on technical role of salt in food is lengthy in description and could be reduced.

Methods – were actual hard copy labels of nutrition information panels checked? Or online sources used? How did you decide what retail outlets to sample in Serbia – can you provide some more information on sampling here.

Thank You. We provided detailed explanation the in the section 2.1. 

“Samples were collected in ten largest chain retail markets in Serbia, comprising 30 different Serbian and imported brands of meat products, which are commonly available in these retails. Sampling laboratory documented all the labels collected for meat products in this exercise (by photos).”

Results and discussion: this starts of with a lot of detail about sources of salt from foods across many different countries before any data from the present result is presented. I understand the journal allows combination of results and discussion but I would recommend in first instance your own study results are clearly presented; then discuss these in the context of the other relevant findings e.g. section 3.1 needs to be removed/reworked elsewhere

Thank you. We did reduce and reorganized the whole results.

The paragraph from line 264, page 6 seems out of context (E.g. methods to determine Na content). Could it not be a simple mention in your limitations section that determining sodium from reported food labels is a limitation in this method and lab based methods such as xxx (summarise) are recommended/needed to confirm findings.

Thank You. Lines from 264 up to 277 deleted.

 Methods – did you take the average of the 2 days of 24hr recall data to collate the information on meat consumption? There is mention of another frequency dietary questionnaire – was this used at all. If so please clarify further what/how the dietary data in the main study was used in your analysis.

For this study, average intake of meat products was assessed using data from two 24HDR only.

Section 3.4 in results is very long, multiple paragraphs describing sodium intakes in different areas. Pick the key areas/population groups to compare your findings to. This section could be reduced significantly

Thank you. We did reduce and reorganized the whole results.

Section 3.5 – while interesting it is incredibly long and ? how much of it is needed to directly talk about the results presented in this study.

Thank you. We did reduce and reorganized the whole results.

(Reviewer 3)

The manus is interesting and important; however, it is very dense, which hinders the reading a lot. I would immediately suggest a mandatory modification and that is to separate the results section from the discussion of the article. This is relevant and will significantly transform the article to a better reading which is not possible in its present form.

Thank you. We did reduce and reorganized the whole results.

 The tables also have a lot of information, and reading will be very beneficial if, instead of putting the salt and sodium values in all the Tables, authors describe only the salt values since sodium derives from it.

Thank you. Since we are referencing WHO (salt) and EFSA (sodium) we prefer to leave both values for a better overview for a reader who seeks deeper info.

 Regarding other methodological issues, I do not find anything relevant that I can add to help the authors. Only the title can be simplified to "Dietary intake of salt from meat products in a Serbian population"

We agreed. Title is now changed.

Round 2

Reviewer 3 Report

The authors addressed the points that were raised in the review.